# Botulinum Neurotoxin A Intravesical Injections in Interstitial Cystitis/Bladder Painful Syndrome: A Systematic Review with Meta-Analysis

**DOI:** 10.3390/toxins11090510

**Published:** 2019-08-30

**Authors:** Antonella Giannantoni, Marilena Gubbiotti, Vittorio Bini

**Affiliations:** 1Department of Medical and Surgical Sciences and Neurosciences, Functional and Surgical Urology Unit, University of Siena, 53100 Siena, Italy; 2Department of Urology, San Donato Hospital, 52100 Arezzo, Italy; 3Serafico Institute of Assisi, Research centre “InVita”, Assisi, 06081 Perugia, Italy; 4Department of Medicine, University of Perugia, 06123 Perugia, Italy

**Keywords:** interstitial cystitis, bladder painful syndrome, chronic pelvic pain, neuropathic pain, botulinum neurotoxin A, onabotulinumtoxinA, abobotulinumtoxin A, intravesical injection

## Abstract

Botulinum neurotoxin A (BoNT/A) appears to be one of the best intravesical treatments for interstitial cystitis/bladder painful syndrome (IC/BPS). We aimed to point out what the evidence is regarding the effects of BoNT/A intravesically injected in patients with IC/BPS. We performed a systematic review of all randomized controlled trials (RCTs) assessing BoNT/A for IC/BPS by using Medline, EMBASE, CINAHL, CENTRAL and MetaRegister of Controlled Trials. Standardized mean differences (SMD) were extracted from the available trials and combined in a meta-analysis applying a random effect model, including heterogeneity of effects. Twelve trials were identified. Significant benefits from BoNT/A injections were detected in: Interstitial Cystitis Symptom Index and Problem Index (ICSI, ICPI) (small to medium effect size: SMD = –0.302; *p* = 0.007 and –0.430, *p* = 0.004, respectively); Visual Analog Scale (VAS) for pain and day-time urinary frequency (medium effect size: SMD = –0.576, *p* < 0.0001 and –0.546, *p* = 0.013, respectively). A great effect size was detected for post-void residual volume (PVR, SMD = 0.728; *p* =0.002) although no clinically relevant in most cases. Great heterogeneity was observed in treatments’ methodologies and symptoms assessment. Overall, BoNT/A intravesical injections significantly improve some of the most relevant symptoms affecting IC/BPS patients.

## 1. Introduction

Chronic pelvic pain syndromes (CPPS) are multifactorial conditions with pain in the pelvic area as the common denominator, potentially sustained by urologic, gynecologic, gastrointestinal, musculoskeletal, neurologic and rheumatologic etiologies, and with dramatic psycho–social concerns [1]. Among CPPS, interstitial cystitis/bladder painful syndrome (IC/BPS) is a clinical syndrome characterized, according to the definition of the European Society for the Study of Interstitial Cystitis/Bladder Painful Syndrome (ESSIC), by the presence of chronic pelvic pain perceived as related to the bladder, with the concomitant presence of at least one other urinary disturbance, i.e., persistent urge to void or increased urinary frequency [2]. The etiology and pathophysiology of IC/BPS are still unknown and treatment modalities vary largely, with many behavioral, pharmacological, mini-invasive and invasive surgical procedures being proposed over time, but all with no satisfactory results [3,4]. Recently, it has been recognized that the disease is not an organ-specific syndrome but a urogenital manifestation of regional or systemic abnormalities characterized by neuropathic pain (NP) [5].

NP is caused by a lesion or disease of the somatosensory system presenting with burning, jabbing or searing sensations and is often associated with allodynia (pain due to a stimulus that does not normally provoke pain), hyperalgesia (increased pain from a stimulus that normally provokes pain) and hyperesthesia or dysesthesia (increased or altered sensitivity to stimulation) [6]. The site of the causative injury in NP can be at peripheral (peripheral nerve, plexus or root) or central level (spinal cord, brainstem or thalamus). In the peripheral NP, damage to the peripheral nervous system leads to irritation of peripheral nerve endings with increased release of nociceptive transmitters (e.g., calcitonin gene-related peptide (CGRP), substance P (SP), glutamate, bradykinin, nerve growth factor). Accumulation of pain transmitters, together with the consequent local inflammation, lowers the sensory threshold of peripheral nerve endings to nociceptive stimuli (peripheral sensitization). Peripheral sensitization determines an increase in arrival of nociceptive signals into the spinal cord and sensitizes spinal sensory neurons (central sensitization). Long-lasting peripheral and central sensitizations lead to chronicity of pain [7]. Both peripheral and central sensitizations have been supposed to play a causative role and a sustaining activity in IC/BPS.

In recent times, botulinum neurotoxins (BoNTs) have been proposed as alternative treatment in IC/BPS patients refractory to conventional therapies [8,9,10]. Laboratory and animal studies have shown that BoNTs can influence pain transmitters in both peripheral and central nervous system (CNS), with a reduction of the behavioral manifestations of pain in treated animals through a variety of mechanisms [11,12]. Also, studies in humans have demonstrated an analgesic effect, particularly for botulinum neurotoxin A (BoNT/A) in pain related disorders, such as chronic migraine (on-label use) [13], or osteoarthritis [14], lower back pain [15], and chronic pelvic pain (off-label use) [16]. Although the neurotoxin A has been initially considered to reduce pain through a simple muscle relaxation, thus providing decreased compression on local blood vessels and nerves [17], recent studies show that the mechanism of action of BoNT/A in pain relief is more complex. Indeed, the antinociceptive effect of BoNT/A has been described in different types of chronic pain not primarily associated with muscular hyperactivity and in different types of neuropathic pain. One explanation may consist of the inhibition of the release of neurotransmitters involved in pain and inflammation at the peripheral level. Indeed, SP, CGRP, glutamate and transient receptor potential vanilloid type 1 (TRPV1) have all been demonstrated to be inhibited by BoNT/A [18,19,20]. Another possible explanation for BoNT/A modulating pain is represented by its retrograde axonal transport to the CNS, by which the neurotoxin may gain access to second-order neurons and play there its activities [21,22,23,24]. With regards to the mechanisms of action of BoNT/A at the level of the urinary bladder, it has been demonstrated that neurotoxin A is able to inhibit the nociceptive, afferent nervous transmission by acting on several receptors and neurotransmitters involved also in neurogenic inflammation [19]. Indeed, BoNT/A has been demonstrated to inhibit the release of SP from cultured dorsal root ganglia neurons [25] and reduces the levels of CGRP in rat trigeminal ganglia cells [26]. In experimental studies in animals, neurotoxin A has been observed to reduce hyperalgesia and TRPV1 expression in rats with neuropathic pain [27], and the expression of cyclooxygenase-2 in a rat model of capsaicin-induced prostatitis [28]. In humans, BoNT/A has been demonstrated to reduce the urinary level of Nerve Growth Factor (NGF) in patients affected by IC/BPS by inhibiting the action of interleukin-1 [29].

Whatever is the prevalent mechanism of action for BoNT/A in modulating chronic pelvic pain syndromes, some non-randomized and randomized controlled studies have been previously produced on the beneficial effects of neurotoxin A in the treatment of patients affected by IC/BPS, somewhat with contradictory results [30,31,32]. In a first systematic review performed by Tirumuru and co-workers on the topic [33], although with a limited evidence, it was suggested that BoNT/A intravesical injections could represent an adequate treatment for patients affected by refractory IC/BPS. A greater evidence supporting the use of BoNTs in bladder pain was then reported by the review of Chiu et al. [19], thus highlighting again the promising role of neurotoxin A in treating patients affected by the disease. In a recent, systematic review with meta-analysis, the role of intravesical BoNT/A injections in IC/BPS patients was examined, and significant improvements in pain, Interstitial Cystitis Symptom Index (ICSI) and Interstitial Cystitis Problem Index (ICPI) [34], urinary frequency and maximum cystometric capacity were obtained by neurotoxin A treatment, although this meta-analysis included a limited number of trials with small sample sizes [30]. The goal of the present systematic review with meta-analysis was to point out what is the actual evidence on the efficacy and safety of BoNT/A in the treatment of patients affected by refractory IC/BPS.

## 2. Results

The initial search allowed the identification of 521 articles. Of these articles, 12 randomized controlled trials (RCTs) met the inclusion criteria and were considered for the meta-analysis [35,36,37,38,39,40,41,42,43,44]. Overall, in the eligible studies, the most frequently assessed outcome was change in ICSI, which was investigated in 10 trials [35,36,38,39,40,41,42,43]; other assessed outcomes, variably reported, were: changes in ICPI in nine trials [35,36,39,43], changes in Visual Analog Scale (VAS) for pain in seven [35,36,39,40,41,43], changes in Likert Scale in two [37,44], changes in daytime urinary frequency in eight [35,36,37,40,41,42], changes in nocturia in eight [35,36,37,40,41,42], changes in maximum flow rate (Qmax) in six [36,39,40,41] and changes in postvoid urinary residual volume (PVR) and in functional bladder capacity in seven [35,36,39,40,41,42]. Table 1 shows the characteristics of the selected trials.

Overall, the total number of patients included was 459, and 419 cases were females. For patients’ inclusion, seven trials [35,37,40,41,42,44] followed the National Institute of Arthritis, Diabetes, Digestive and Kidney Diseases (NIDDK) criteria [45], three adhered to the European Society for the Study of Interstitial Cystitis (ESSIC) criteria [36,43] and two other trials considered clinical and/or cystoscopic findings [38,39]. In the majority of the trials, all the included patients had been refractory to previous treatments [35,36,37,38,40,41,42,43,44]. In two studies no description about previous treatments’ modalities was reported [39,44]. Finally, one study was aimed to include only patients affected by IC/BPS presenting with Hunner lesions [39].

As noted, a large variability in the neurotoxin dosages, injection modalities and sites of injections has been found in the selected RCTs. Indeed, onabotulinumtoxinA (Onabot/A) was used in 11 trials (Table 1); different dosages of the neurotoxin have been used, varying from 50 to 300 U; sites of injections were: the trigone [35,39,43], the whole bladder wall [44], out of the trigone [40,41,42], or the bladder neck [38]. Only in one study abobotulinumtoxin A (Abobot/A) 500 U has been used as the active treatment [42]. Furthermore, also deep of injection into the bladder wall varied among the selected trials, with the neurotoxin being injected intradetrusorially or sub-urothelially [35,40,41,43]. In some cases, neurotoxin administration was performed under general anesthesia [35,41,43], spinal anesthesia [39] or local anesthesia [37], and in the remaining studies, types of anesthesia have not been described [36,38,40,44]. A large variability we also found in the agents used in control groups, with hydrodistension, normal saline, normal saline plus hydrodistension, bacillus of Calmette–Guerin (BCG), lipotoxin and pentosan polysulfate sodium (PPS) instillations being used in different studies (Table 1). In addition, in one study the control group consisted of patients who received Onabot/A delayed injections as compared to patients in the active group, who received immediate neurotoxin A administration [35].

### 2.1. Outcomes

#### 2.1.1. Effect Size of Standardized Mean Difference on Interstitial Cystitis Symptom Index

The 10 trials reporting on ICSI showed an effect size ranging from small to medium, with a SMD of −0.302; *p* = 0.007. Thus, BoNT/A intravesical injection was significantly more effective than control agents in improving the means score of ICSI with a small to medium effect size of SMD (Table 2, Figure 1).

In four of these studies, onabot/A different units (from 50 to 300 U) were compared with normal saline [35,38,40,43]; in three studies onabot/A different units were compared with bladder hydrodistension [39,40,42]; in one study, 100 U onabot/A immediate injection was compared to delayed neurotoxin injection [35]; in another study, onabot/A 300 U were compared to intravesical BCG [37], and finally, in one study onabot/A 200 U were compared to lipotoxin intravesical instillations [36]. Only one study investigated the effects of abobot/A 500 U plus hydrodistension in comparison with normal saline with the addition of hydrodistension [42].

#### 2.1.2. Effect Size of Standardized Mean Difference on Interstitial Cystitis Problem Index

Nine of the selected RCTs reported the results related to changes in ICPI (Table 3, Figure 2).

Overall, the effect size of SMD ranged from small to medium (SMD = –0.430, *p* = 0.004) [35,36,39,40,41,42,43]. Thus, treatment with BoNT/A intravesical injection was significantly more effective than control agents in improving ICPI, with a small to medium effect size of SMD. Again, in these studies onabot/A different dosages were compared to normal saline, hydrodistension, lipotoxin or BCG intravesical instillations. In one trial, abobot/A plus hydrodistension was compared to normal saline plus hydrodistension [35].

#### 2.1.3. Effect Size of Standardized Mean Difference on Visual Analog Scale or Likert Scale

Nine trials examined the effects of onabot/A injections on VAS or Likert scale. In these studies, the effect size of SMD was found to be as medium (SMD = –0.576, *p* < 0.0001), with BoNT/A significantly improving VAS for pain or Likert scale as compared to control agents [35,36,37,39,40,41,43] (Table 4, Figure 3).

#### 2.1.4. Effect Size of Standardized Mean Difference on Daytime Urinary Frequency

Eight trials examined the effects of the neurotoxin on day-time urinary frequency and also in these studies BoNT/A intravesical injection significantly improved day-time urinary frequency as compared to control agents, with a medium effect size of SMD of –0.546, (*p* = 0.013) [35,36,37,40,41,42], (Table 5, Figure 4).

#### 2.1.5. Effect Size of Standardized Mean Difference on Nocturia

Changes in nocturia were examined in eight trials [35,36,37,40,41,42]. From these studies, no significant effect size of BoNT/A intravesical injections was observed as compared to control agents (SMD = –0.183, *p* = 0.278). Thus BoNT/A did not significantly improve nocturia as compared to control agents, in patients affected by IC/BPS (Table 6, Figure 5).

#### 2.1.6. Effect Size of Standardized Mean Difference on Functional Bladder Capacity

Changes in bladder capacity were investigated in seven trials [35,36,40,41,42]. The effect size of SDM was not significant (SMD = 0.194, *p* = 0.227), indicating no significant effect of the neurotoxin A injection in improving this outcome, as compared to control agents (Table 7, Figure 6).

#### 2.1.7. Effect Size of Standardized Mean Difference on Maximum Flow Rate

Five trials examined the effects of the neurotoxin on maximum flow rate [36,39,40,41], and also in these trials no significant effect size was observed (SMD = –0.017; *p* = 0.948), (Table 8, Figure 7).

#### 2.1.8. Effect Size of Standardized Mean Difference on Post Void Urinary Residual Volume (PVR)

Seven trials investigated the effect of the neurotoxin on PVR [35,36,39,40,41]. As a result, a great effect size of SMD was detected (SMD = 0.728; *p* =0.002). As expected, BoNT/A intravesical injection significantly increase PVR as compared to control agents (Table 9, Figure 8).

### 2.2. Side Effects

Side effects have been described in 10 of 12 trials and, when reported, the detection time of complications largely varied from case to case. Akyama and co-workers did not observe any grade III surgical complications according to the Clavien classification [35]. In the same study, transient hematuria was noted in one patient and symptomatic urinary tract infections (UTIs) in two [35]. Kuo and Chancellor observed hematuria in two cases who received onabot/A 200 U; in addition, the authors noted dysuria and large PVR in seven and three patients treated with onabot/A 200 and 100 U, respectively, a condition which lasted for two weeks after the injection treatment [40]. In the same study, urinary retention was detected in two cases who underwent onabot/A 200 U, which resolved after six months [40]. Dysuria was the most frequently observed side effect in the selected RCTs, with 36 and 14 patients being affected after the active or the control treatment, respectively. In the study of Kuo and co-workers of 2016, dysuria was detected in 13 patients in the neurotoxin group and in one case in the control group [41]. In the same study, one patient treated with the neurotoxin A presented with simultaneous UTI and urinary retention, and two additional cases had transient hematuria [41]. UTI was the most frequent complication in the study of Manning and co-workers, affecting seven patients treated with abobot/A and five in the control group [42]. The authors considered UTI a confounding factor for the analysis of their results. UTIs were also reported in the study of Pinto et al. with three patients having UTIs in the onabot/A group and two in the control group [43]. Overall, 16 patients in the active group and 11 in the control group presented with UTIs after treatment, and the difference was not significant. Only one patient presented with dysuria in the study of Chuang and Kuo [36], while Gottsch and co-workers did not observe any complication from the neurotoxin A injections [38]. The description of side effects was not reported in the studies of Kasyan et al. and Taha-Rashed et al. [39,44]. The description of side effects has been reported in Table 10.

## 3. Discussion

Overall, in the present review with meta-analysis of RCTs on the use of BoNT/A injections in the treatment of refractory patients affected by IC/BPS, we were able to detect a medium effect size of SMD for changes in VAS for pain and changes in daytime urinary frequency, and a small to medium effect size of SMD for changes in ICSI and ICPI. No effect size of SMD was identified with regards to changes in nocturia and in functional bladder capacity, nor in maximum flow rate. A great effect size of SMD was detected only for changes in PVR, which is, as expected, a possible complication of the neurotoxin injection into the bladder wall and it represents a side effect.

Thus, in the present review with meta-analysis, BoNT/A intravesical administration has been shown to be able to significantly improve pain and daily frequency of urination in refractory IC/BPS patients, a remarkable effect in these patients so afflicted by their painful condition. Indeed, the detected effects indicates a moderate effectiveness of BoNT/A in decreasing bladder pain and daily frequency of urination, in comparison with the control agents. Significant ameliorations in ICSI and ICPI were also identified here, which represents other important effects of the neurotoxin A administration, considering that ICSI and ICPI questionnaires are reliable and valid standardized tools to assess urinary symptoms and pain in patients with IC/BPS. Nevertheless, the effect size of SDM for both ICSI and ICPI ranged from small to medium. Unfortunately, no effects have been identified on changes in functional bladder capacity and nocturia. It is well known that nocturia is one of the most difficult urinary symptom to be treated, possibly due to concomitant, different etiologies in the same subjects [46]. Functional bladder capacity was an outcome assessed in seven of the selected RCTs and surprisingly, a great reduction in functional bladder capacity after the active agent administration has been described in one of these studies [35]. This presumably prevented the obtainment of the expected result of a significant improvement in bladder capacity, which is usually observed after BoNT/A injections in patients with overactive bladder [47]. PVR was been found to significantly increase after treatment, with a great effect size of SMD, in the majority of the examined studies, but it appeared of poor clinical relevance, and only four cases presented with urinary retention after the administration of the active agent [40,41]. In line with the detection of limited increases in PVR, Qmax was not affected by the neurotoxin injections, as no effect size of SMD was identified. With regards to the rate of UTIs, this was low (3.8%) after the BoNT/A injection, and also the 7.8% rate of transient dysuria detected in patients treated with the neurotoxin can be considered as low. Thus BoNT/A intravesical injections in IC/BPS patients appeared to be safe and without any serious or systemic complication. The significant amelioration we found in VAS for pain/Likert scale, and in ICSI and ICPI after BoNT/A intravesical injections in IC/BPS patients confirms the previously reported significant benefits of BoNT/A by Wang and co-workers in their meta-analysis [30], but we included a larger number of RCTs and used a different methodology in the analysis of the results. Indeed, the use of SMD as we performed, serves as an easy way to judge the magnitude of the treatment’s effect, and the real effect can be better understood and is clinically more interpretable. Zhang and co-workers in their network meta-analysis on the intravesical treatment of IC/BPS found that BoNT/A had the highest probability to be the best therapy for the affected patients in comparison with other intravesical treatments, as BCG, PPS, lidocaine, chondroitin sulfate and resiniferatoxin [32]. In another review with meta-analysis performed on the same topic, Shim and co-workers also used the SMD for the measurement of the effects, and found significant improvements in VAS for pain, ICPI and urinary frequency in patients treated with BoNT/A vs. controls [31]. No significant effects were identified in that review on other outcomes (ICSI, nocturia, functional bladder capacity), but only five RCTs have been included in their analysis [31]. Presumably, the larger number of trials with a higher number of treated patients included in our review with meta-analysis, as compared to those of Shim et al. allowed us to detect the significant improvement also in ICSI after BoNT/A intravesical injections. Tirumuru et al. included in their systematic review only three RCTs and seven prospective studies, with a limited number of treated IC/BPS patients with the neurotoxin A [33]. They found improvement in symptoms in two trials and in six prospective studies, but meta-analysis was not performed due to heterogeneity of included outcomes and it was stated that no solid conclusions could be derived from their review on the effectiveness of BoNT/A intravesical injection in the treatment of IC/BPS [33].

When considering the quality of the selected trials, six trials in our review showed a high quality score (Jadad score (JS) = 5), [36,40,41,43]. A JS of three was obtained by three trials [35,38,42], and a JS of two by three other RCTs [37,39,44]. Indeed, two studies with a two JS were published as abstracts [39,44], and no full publications have been followed until now from the same authors. Of note, only a JS scores ≥ 3 indicates a good methodological quality of the study, thus in the present review with meta-analysis, 9/12 RCTs were found to be of good quality.

With the exclusion of the following outcomes: ICSI and VAS/Likert scale, which are two very important and reliable tools in evaluating the response to treatment, the present meta-analysis shows a great heterogeneity that could be attributed to differences in treatment’s methodology, selected outcomes, symptoms assessment and length of follow-up along the selected trials. If on the one hand, performing a meta-analysis could have been inappropriate due to the heterogeneity identified, combining the data of each individual outcome in the respective meta-analysis and summarizing the pooled effect sizes however helped us bring order to the data coming from the existing literature, and allowed us to point to the evidence of BoNT/A intradetrusorial treatment in IC/BPS.

It should be argued that a greater consistency about the neurotoxin dosages to be used in this specific patients’ population, sites, depth and number of injections into the bladder, the eventual use of hydrodistension (how and when), could have allowed us to reach more meaningful results in order to definitely assess the benefits of BoNT/A in controlling pain and urinary symptoms in patients affected by IC/BPS.

## 4. Materials and Methods

### 4.1. Literature Search

This meta-analysis was conducted according to predefined guidelines provided by the standard PRISMA protocol [48]. We aimed to critically review and synthesize data from the current therapeutic use of BoNT/A in the treatment of IC/BPS, to quantify the effect size from randomized controlled trials (RCTs). A systematic review of the literature to identify RCTs on the treatment of IC/BPS published between January 2000 and March 2019 was performed. We conducted Medline, EMBASE, CINAHL, CENTRAL and MetaRegister of Controlled Trials searches using the search terms: bladder painful syndrome, interstitial cystitis, hypersensitive bladder, bladder hypersensitive disorders. In the searches we also included the following terms: botulinum neurotoxin A, onabotulinumtoxinA, botox, abobotulinumtoxinA, dysport, incobotulinumtoxinA, xeomin, neurotoxin A, botulinum toxin A intravesical injections, and we associated them to the previously indicated search terms: bladder painful syndrome, interstitial cystitis, hypersensitive bladder, bladder hypersensitive disorders.

### 4.2. Inclusion and Exclusion Criteria

Only articles in the English language published on the use of botulinum toxin A intravesical injections to treat patients affected by IC/BPS were included in the search. Then we identified all original researches in the form of RCTs, and we excluded case reports and non-human studies. Review articles have been taken into account for the comparison of final data. The references of review articles were also surveyed to identify any potentially missed articles. Antonella Giannantoni and Marilena Gubbiotti reviewed each title and, if unclear, the full article applying the inclusion and exclusion criteria.

### 4.3. Assessment of Results

Data were extracted from all RCTs as standardized mean differences (SMD), as previously described [49]. The use of SMD allows the measurement of the effect for each interventional trial on a similar metric, and it derives this by dividing the difference in mean outcome between two groups, taking into account the pooled standard deviation of the measurement. The obtained effect size scores can be negative or positive, no change is indicated by a score of 0, and the outcome is measured in standard deviation units. According to Cohen [50], a SMD of 0.2 standard deviation units expresses a small difference between the interventional groups; a SMD of 0.5 indicates a medium difference, and a value of 0.8 is indicative of a great difference [50]. The so extracted SMDs were then integrated in a meta-analysis with a random-effect model and with also the assessment of the heterogeneity of the effects by means of the Cochran Q test and I^2^ statistics [51,52]. Heterogeneity was considered significant in cases of *p* < 0.10 and *I*^2^ > 50%. Publication bias was measured by the Begg and Egger tests [52,53]. Finally, the Jadad score (Js) was also used to assess the quality of the included trials, with an overall score of three being representative of a high-quality study [54]. All measurements were performed using Stats Direct statistical software v.2.7.2( Stats Direct Ltd Merseyside, UK, 2008).

We previously analyzed the outcomes assessed in each randomized, controlled study we found on the use of BoNT/A intravesical injections in patients affected by IC/BPS. Overall, 10 RCTs were identified [35,36,37,38,39,40,41,42,43,44]. Two out of the ten studies we found, reported the results related to two different trials, with patients receiving different BoNT/A dosages in one trial [40], or different control agents in the other trial [36]. Thus, the eligible, considered trials were 12. In the selected studies, in order to investigate the benefit coming from the neurotoxin A injection or from the control agent administration, the assessed outcomes were classified as primary or secondary outcomes, i.e., ICSI and ICPI [34], VAS for pain or Likert scale, day-time urinary frequency, nocturia, bladder capacity (in the form of cystometric bladder capacity or functional bladder capacity). Changes in other outcomes, as Qmax and PVR have been considered as side effects.

## 5. Conclusions

BoNT/A intravesical injections are able to significantly improve some of the most remarkable symptoms, namely pain and frequency of urination, in patients affected by IC/BPS, and without serious and/or long-lasting side effects. However, due to the still limited evidence, more large-scale, RCTs with higher consistency in methodologies are needed.

## Figures and Tables

**Figure 1 toxins-11-00510-f001:**
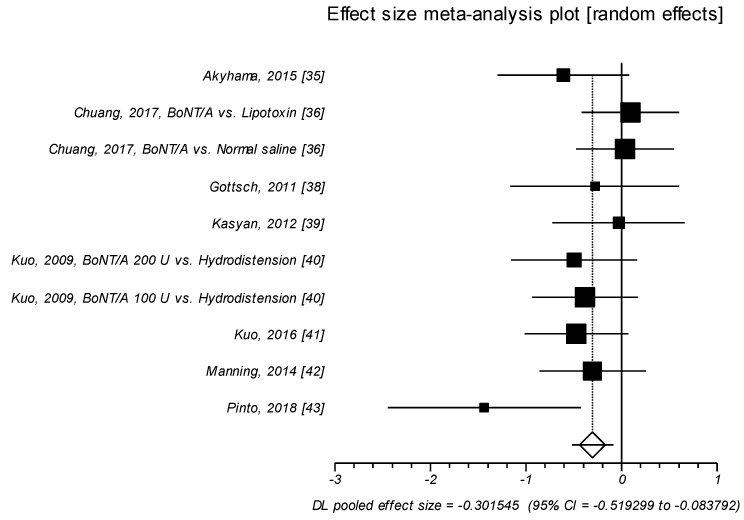
ICSI. Effect size meta-analysis plot (random effects). Cochran Q test: *p* = 0.279; *I*^2^ = 17.8%.

**Figure 2 toxins-11-00510-f002:**
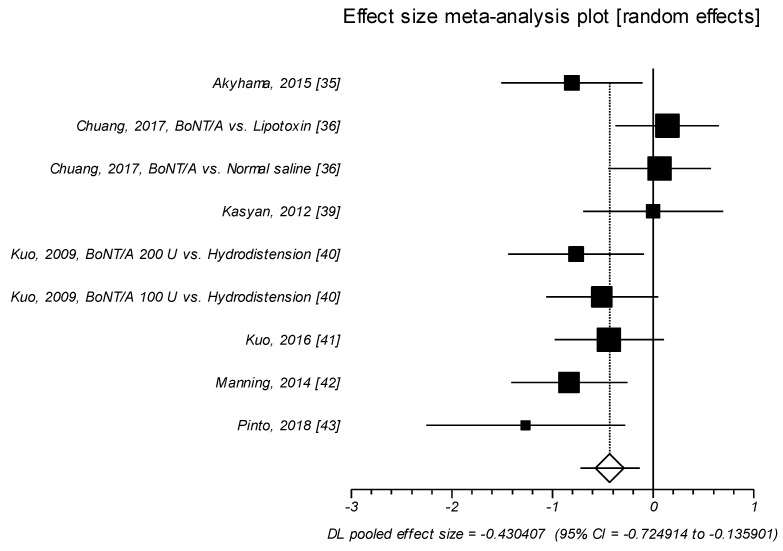
ICPI. Effect size meta-analysis plot (random effects). Cochran Q test: *p* = 0.036; *I*^2^ = 51.3%.

**Figure 3 toxins-11-00510-f003:**
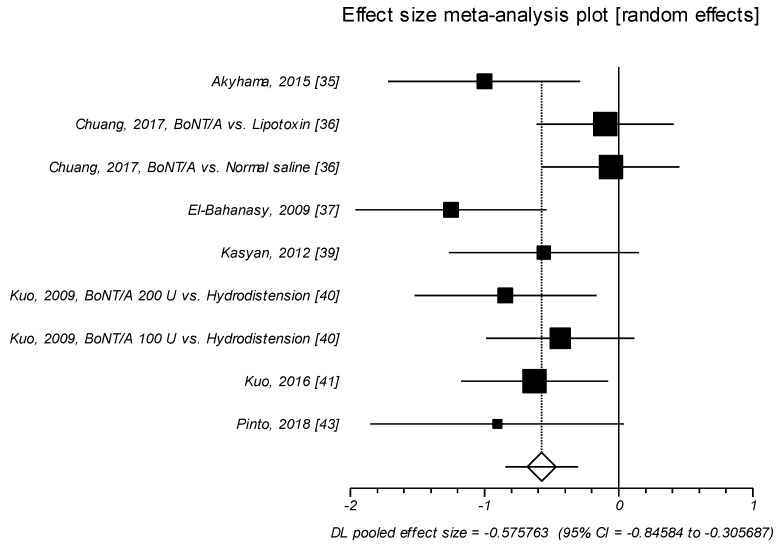
VAS/Likert Scale. Effect size meta-analysis plot (random effects). Cochran Q test: *p* = 0.105; *I*^2^ = 39.3%.

**Figure 4 toxins-11-00510-f004:**
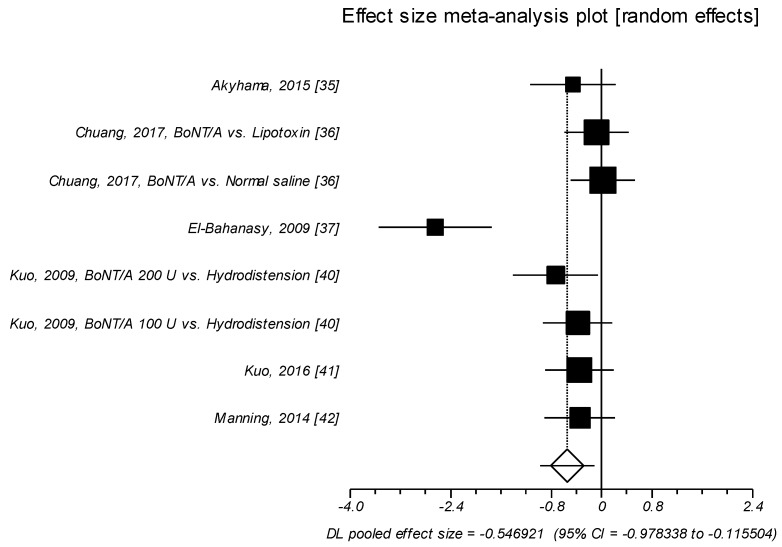
Day-time urinary frequency. Effect size meta-analysis plot (random effects). Cochran Q test: *p* = 0.0001; *I*^2^ = 76.1%.

**Figure 5 toxins-11-00510-f005:**
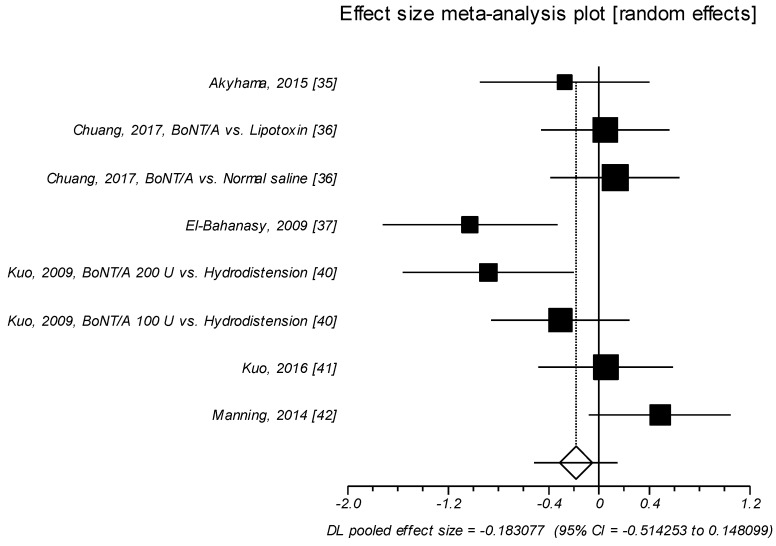
Nocturia. Effect size meta-analysis plot (random effects). Cochran Q test: *p* = 0.012; *I*^2^ = 61.3%.

**Figure 6 toxins-11-00510-f006:**
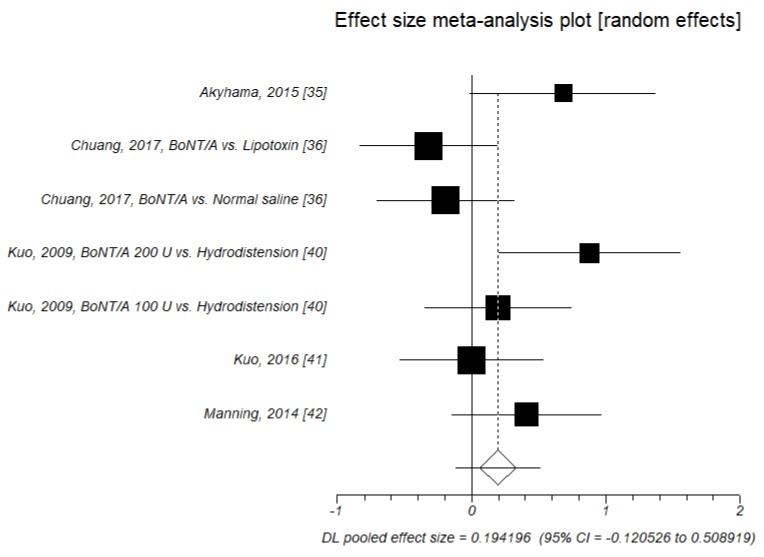
Functional bladder capacity. Effect size meta-analysis plot (random effects). Cochran Q test: *p* = 0.047; *I*^2^ = 53%.

**Figure 7 toxins-11-00510-f007:**
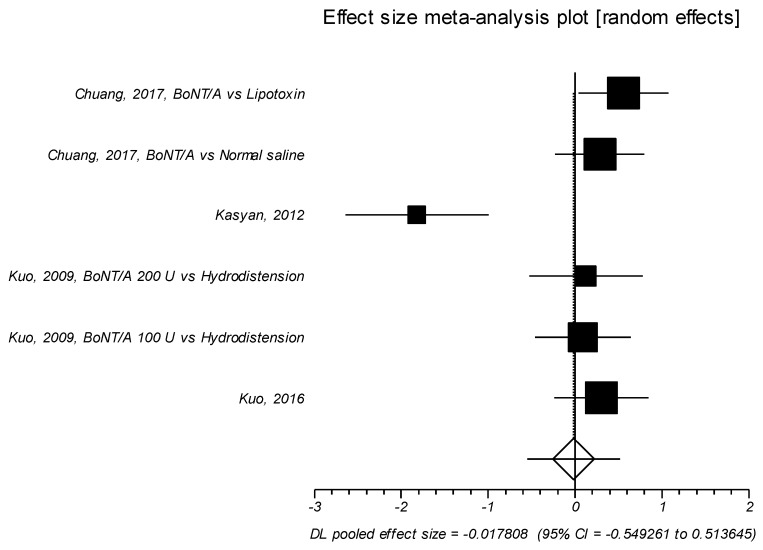
Maximum flow rate. Effect size meta-analysis plot (random effects). Cochran Q test: *p* = 0.0002; *I*^2^ = 79.8%.

**Figure 8 toxins-11-00510-f008:**
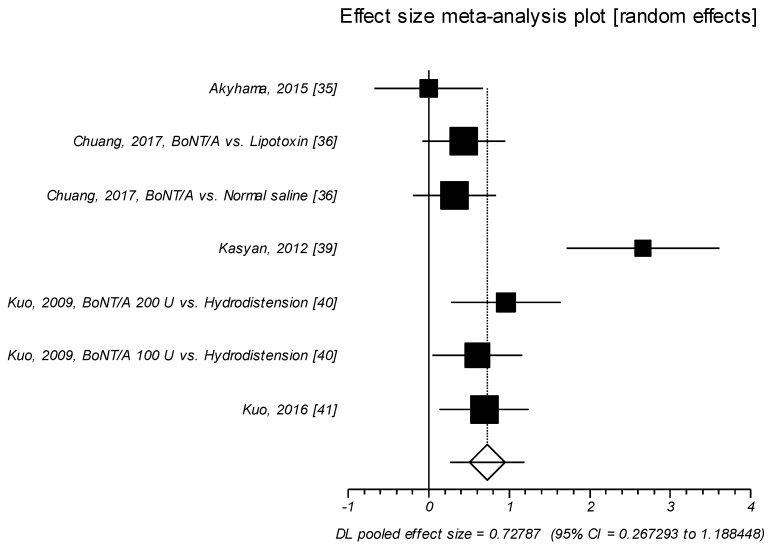
PVR. Effect size meta-analysis (random effects). Cochran Q test: *p* = 0.0006; *I*^2^ = 74.7%.

**Table 1 toxins-11-00510-t001:** Characteristics of randomized controlled trials (RCTs) on the use of botulinum neurotoxin A intravesical injections in the treatment of patients affected by interstitial cystitis/bladder painful syndrome (IC/BPS).

Author, Year, Treatment	Js	Diagnostic Criteria	Design	Total No. of Pts	Females/ Males	Patients’ Age, (y) Mean ± SD or Mean (min-ax)	Disease Duration (y) Mean ± SD	Active Agent	Control Agent	No. of Pts atBaseline Active Agent /Control	No. of Pts at Follow-Up Active Agent /Control	Follow-up, months
**Akyhama, 2015, [35]**	3	NIDDK	RCT	34	26/8	64.9 ± 13.7	6.6 ± 4.4	Onab 100 U i.ve, immediate injection	Onab 100 U i.ve 1-month delayed injection	18/16	18/16	1
**Chuang, 2017, [36]** BoNT/A vs. Lipotoxin	5	ESSIC	RCT	90	80/10	52.5 ± 4.2	7.2 ± 6.0	Onab 200 U i.ve	Lipotoxin i.ve	28/31	28/31	1
**Chuang 2017, [36]** BoNT/A vs. Normal Saline	5	ESSIC	RCT	90	80/10	52.5 ± 4.2	7.2 ± 6.0	Onab 200 U i.ve	Normal saline i.ve	28/31	28/31	1
**Gottsch, 2011, [38]**	3	Clinical	RCT	20	20/0	45.8 (22–62)	16.5 (2–30)	Onab 50 U peri-urethral	Normal saline i.ve	9/11	9/11	3
**El-Bahanasy, 2009, [37]**	2	NIDDK	RCT	36	36/0	NA	NA	Onab 300 U i.ve	BCG i.ve	18/18	18/16 (at 1 month)	5.5 vs. 5.75
**Kasyan, 2012, [39]**	2	Clinical, cystoscopic	RCT	32	32/0	NA	NA	Onab 100 U i.ve	Hydrodistension	15/17	15/17	3
**Kuo, 2009, [40]** BoNT/A 200 U vs. Hydrodistension	5	NIDDK	RCT	38	31/7	49.1 (26–83)	8 ± 5	Onab 200 U i.ve	Hydrodistension	15/23	15/23	3
**Kuo, 2009, [40]** BoNT/A 100 U vs. Hydrodistension	5	NIDDK	RCT	52	45/7	50.1 (26–83)	8 ± 5	Onab 100 U i.ve	Hydrodistension	29/23	29/23	3
**Kuo, 2016, [41]**	5	NIDDK	RCT	60	52/8	51.5 (20–82)	5 ± 2.8	Onab 100 U i.ve	Normal saline i.ve	40/20	40/20	2
**Manning, 2014, [42]**	3	NIDDK	RCT	50	50/0	53.5	13.5 ± 6.75	Abob 500 U i.ve + hydrodistension	Normal saline i.ve + hydrodistension	25/25	25/25	3
**Pinto, 2018, [43]**	5	ESSIC	RCT	19	19/0	45.8 ± 10.5	NA	Onab 100 U i.ve	Normal saline i.ve	11/10	10/9	3
**Taha-Rasheed,** **2010, [44]**	2	NIDDK	RCT	28	28/0	NA	NA	BoNT/A 300 U i.ve	PPS	14/14	14/14	4.75 vs. 5.25

Abbreviations. BoNT/A: Botulinum neurotoxin A; Js: Jadad score; NIDDK: National Institute of Arthritis, Diabetes, Digestive and Kidney Diseases; ESSIC: European Society for the Study of Interstitial Cystitis/Bladder Painful Syndrome; Onab: onabotulinumtoxinA; Abob: abobotulinumtoxinA; i.ve: intravesical; BCG: bacillus of Calmette–Guerin; PPS: pentosan polysulfate sodium.

**Table 2 toxins-11-00510-t002:** RCTs on the use of botulinum neurotoxin A intravesical injections in the treatment of patients affected by IC/BPS, assessing changes in Interstitial Cystitis Symptom Index (ICSI).

ICSI	Active Agent, Baseline			Control Agent, Baseline			Active Agent, f-up			Control Agent, f-up		
Study, Treatment	Mean	SD	No. of pts	Mean	SD	No. of pts	Mean	SD	No. of pts	Mean	SD	No. of pts
Akyhama, 2015, [35]	14.2	3.6	18	12.6	4.3	16	−3.1 *	3.9	18	−0.8 *	3.4	16
Chuang, 2017, BoNT/A vs. Lipotoxin [36]	11.9	3.93	28	12.4	4.12	31	8.29	3.68	28	8.42	4.35	31
Chuang, 2017, BoNT/A vs. Normal saline [36]	11.9	3.93	28	11.2	3.4	31	8.29	3.68	28	7.45	3.66	31
Gottsch, 2011, [38]	35.2	3.9	9	29.6	8	11	31.3	7.5	9	27.7	7.3	11
Kasyan, 2012, [39]	14.5	2.3	15	13.8	3.7	17	9.4	2.9	15	8.8	3.3	17
Kuo, 2009, BoNT/A 200 U vs. Hydrodistension [40]	13.9	2.53	15	12.8	3.41	23	8.9	5.58	15	9.87	4.85	23
Kuo, 2009, BoNT/A 100 U vs. Hydrodistension [40]	12.5	2.15	29	12.8	3.41	23	8.17	4.06	29	9.87	4.85	23
Kuo, 2016, [41]	13.3	3.8	40	12.3	3.9	20	8.8	4.2	40	9.8	5.1	20
Manning, 2014, [42]	13.2	2.6	26	13.9	2.8	27	10.5	4.4	25	12.3	4.5	25
Pinto, 2018, [43]	15.7	3.3	11	13.6	2.3	10	−9 *	4.7	10	−2 *	4.6	9

Data are expressed as mean ± SD for the active and the control agents, at both baseline and follow-up (f-up). * Value has been reported as difference from baseline.

**Table 3 toxins-11-00510-t003:** RCTs on the use of botulinum neurotoxin A intravesical injections in the treatment of patients affected by IC/BPS, assessing changes in Interstitial Cystitis Problem Index (ICPI).

ICPI	Active Agent, Baseline			Control Agent, Baseline			Active Agent, f-up			Control Agent, f-up		
Study, Treatment	Mean	SD	No. of pts	Mean	SD	No. of pts	Mean	SD	No. of pts	Mean	SD	No. of pts
Akyhama, 2015 [35]	12.1	2.5	18	10.6	3.2	16	−2.9 *	3.6	18	−0.1 *	3.1	16
Chuang, 2017, BoNT/A vs. Lipotoxin [36]	11.4	3.92	28	11.8	3.9	31	8.64	4.3	28	8.42	5.44	31
Chuang, 2017, BoNT/A vs. Normal saline [36]	11.4	3.92	28	11	4.14	31	8.64	4.3	28	7.97	4.42	31
Kasyan, 2012, [39]	12.4	2.4	15	11.9	3.1	17	7.3	2.1	15	6.8	2.5	17
Kuo, 2009, BoNT/A 200 U vs. Hydrodistension [40]	12.3	1.4	15	11.1	2.6	23	7.13	4.52	15	8.57	4.59	23
Kuo, 2009, BoNT/A 100 U vs. Hydrodistension [40]	11.1	2.05	29	11.1	2.6	23	6.93	3.58	29	8.57	4.59	23
Kuo, 2016, [41]	12.6	3	40	11.7	3.8	20	7.6	4.2	40	8.4	4.8	20
Manning, 2014, [42]	13.6	2.54	25	13.7	2.66	25	9.9	4	25	12.8	4	25
Pinto, 2018, [43]	13.3	1.5	10	12.2	2.2	9	−7.1 *	4.6	10	−1 *	4.6	9

Data are expressed as mean ± SD for the active and the control agents, at both baseline and follow-up (f-up). * Value has been reported as difference from baseline.

**Table 4 toxins-11-00510-t004:** RCTs on the use of botulinum neurotoxin A intravesical injections in the treatment of patients affected by IC/BPS, assessing changes in Visual Analog Scale (VAS)/Likert scale.

VAS/Likert Scale	Active Agent, Baseline			Control Agent, Baseline			Active Agent, f-up			Control Agent, f-up		
Study, Treatment	Mean	SD	No. of pts	Mean	SD	No. of pts	Mean	SD	No. of pts	Mean	SD	No. of pts
Akyhama, 2015 [35]	7.3	1.7	18	6.3	2.5	16	−2.2 *	2	18	−0.1 *	2.1	16
Chuang, 2017, BoNT/A vs. Lipotoxin [36]	4.5	3.22	28	4.84	2.34	31	2.57	2.54	28	3.19	2.71	31
Chuang, 2017, BoNT/A vs. Normal saline [36]	4.5	3.22	28	4.32	2.65	31	2.57	2.54	28	2.55	2.23	31
El-Bahanasy, 2009 [37]	5.8	1.39	18	5.4	1.23	18	0.22	0.43	18	1.06	0.77	18
Kasyan, 2012 [39]	9.3	0.9	15	8.7	1.2	17	5.8	2.4	15	6.1	1.8	17
Kuo, 2009, BoNT/A 200 U vs. Hydrodistension [40]	5.47	2.1	15	4.3	2.6	23	2.47	2.1	15	3.52	3.07	23
Kuo, 2009, BoNT/A 100 U vs. Hydrodistension [40]	4.83	2.21	29	4.3	2.6	23	2.97	1.99	29	3.52	3.07	23
Kuo, 2016 [41]	5.3	2.6	40	3.7	2.9	20	2.7	2.7	40	2.8	2.5	20
Pinto, 2018 [43]	6.8	1.2	11	6.8	0.8	10	−3.8 *	2.5	10	−1.6 *	2.1	9

Data are expressed as mean ± SD for the active and the control agents, at both baseline and follow-up (f-up). * Value has been reported as difference from baseline.

**Table 5 toxins-11-00510-t005:** RCTs on the use of botulinum neurotoxin A intravesical injections in the treatment of patients affected by IC/BPS, assessing changes in day-time urinary frequency.

Daytime Urinary Frequency	Active Agent, Baseline			Control Agent, Baseline			Active Agent, f-up			Control Agent, f-up		
Study, Treatment	Mean	SD	No. of pts	Mean	SD	No. of pts	Mean	SD	No. of pts	Mean	SD	No. of pts
Akyhama, 2015 [35]	18.6	8	18	22.6	13.1	16	−2.9 *	5.1	18	−1 *	2.5	16
Chuang, 2017, BoNT/A vs. Lipotoxin [36]	12.8	5.2	28	14.3	7.09	31	11.5	4.82	28	13.5	7.77	31
Chuang, 2017, BoNT/A vs. Normal saline [36]	12.8	5.2	28	12.9	6.6	31	11.5	4.82	28	11.5	4.82	31
El-Bahanasy, 2009 [37]	16.8	2.6	18	16.7	3.2	18	5.3	1.14	18	11.5	2.34	18
Kuo, 2009, BoNT/A 200 U vs. Hydrodistension [40]	14.2	5.44	15	11.6	4.36	23	9.4	3.22	15	9.96	3.97	23
Kuo, 2009, BoNT/A 100 U vs. Hydrodistension [40]	13	4.69	29	11.6	4.36	23	9.72	4.03	29	9.96	3.97	23
Kuo, 2016 [41]	14.3	6	40	13.7	9.1	20	10.5	5.1	40	12.4	9.6	20
Manning, 2014 [42]	13.5	7.1	25	12.5	5.4	25	10.4	5.8	25	11.4	4.4	25

Data are expressed as mean ± SD for the active and the control agents, at both baseline and follow-up (f-up). * Value has been reported as difference from baseline.

**Table 6 toxins-11-00510-t006:** RCTs on the use of botulinum neurotoxin A intravesical injections in the treatment of patients affected by IC/BPS, assessing changes in nocturia.

Nocturia	Active Agent, Baseline			Control Agent, Baseline			Active Agent, f-up			Control Agent, f-up		
Study, Treatment	Mean	SD	No. of pts	Mean	SD	No. of pts	Mean	SD	No. of pts	Mean	SD	No. of pts
Akyhama, 2015 [35]	4.2	3.1	18	5.1	4.8	16	−0.6 *	2.4	18	−0.1 *	0.6	16
Chuang, 2017, BoNT/A vs. Lipotoxin [36]	2.96	1.48	28	3.46	2.35	31	2.74	1.58	28	3.13	3.04	31
Chuang, 2017, BoNT/A vs. Normal saline [36]	2.96	1.48	28	3.15	1.65	31	2.74	1.58	28	2.71	2.02	31
El-Bahanasy, 2009 [37]	6.3	1.8	18	6.06	6.06	18	0.28	0.48	18	2.78	1.08	18
Kuo, 2009, BoNT/A 200 U vs. Hydrodistension [40]	6.33	6.96	15	3.7	2.03	23	3.13	2.47	15	3.52	2.15	23
Kuo, 2009, BoNT/A 100 U vs. Hydrodistension [40]	3.41	2.16	29	3.7	2.03	23	2.59	1.97	29	3.52	2.15	23
Kuo, 2016 [41]	3.5	1.3	40	4.3	2.6	20	2.8	1.3	40	3.5	2.3	20
Manning, 2014 [42]	3.2	1.6	25	3.2	2.6	25	3.3	2.2	25	2.3	1.7	25

Data are expressed as mean ± SD for the active and the control agents, at both baseline and follow-up (f-up). * Value has been reported as difference from baseline.

**Table 7 toxins-11-00510-t007:** RCTs on the use of botulinum neurotoxin A intravesical injections in the treatment of patients affected by IC/BPS, assessing changes in functional bladder capacity.

Bladder Capacity	Active Agent, Baseline			Control Agent, Baseline			Active Agent, f-up			Control Agent, f-up		
Study, Treatment	Mean	SD	No. of pts	Mean	SD	No. of pts	Mean	SD	No. of pts	Mean	SD	No. of pts
Akyhama, 2015 [35]	201.9	131.6	18	145.3	73.3	16	35 *	78.5	18	−10 *	43.4	16
Chuang, 2017, BoNT/A vs. Lipotoxin [36]	309	144	28	262	114	31	315	118	28	307	110	31
Chuang, 2017, BoNT/A vs. Normal saline [36]	309	144	28	298	134	31	315	118	28	332	169	31
Kuo, 2009, BoNT/A 200 U vs. Hydrodistension [40]	113.9	58	15	134	72.4	23	190.8	80.6	15	145.5	77.4	23
Kuo, 2009, BoNT/A 100 U vs. Hydrodistension [40]	161	97.4	29	134	72.4	23	189	78.8	29	145.5	77.4	23
Kuo, 2016 [41]	158.1	97.7	40	127.5	57.3	20	219.6	103.6	40	189	99.4	20
Manning, 2014 [42]	242	166	25	233	96	25	273	152	25	210	84	25

Results are expressed in mL, as mean ± SD for the active and the control agents, at both baseline and follow-up (f-up). * Value has been reported as difference from baseline.

**Table 8 toxins-11-00510-t008:** RCTs on the use of botulinum neurotoxin A intravesical injections in the treatment of patients affected by IC/BPS, assessing changes in maximum flow rate.

Maximum Flow Rate	Active Agent, Baseline			Control Agent, Baseline			active Agent, f-up			Control Agent, f-up		
Study, Treatment	Mean	SD	No. of pts	Mean	SD	No. of pts	Mean	SD	No. of pts	Mean	SD	No. of pts
Chuang, 2017, BoNT/A vs. Lipotoxin [36]	13.7	7.6	28	14.4	8.07	31	21.2	9.31	28	17.1	8.97	31
Chuang, 2017, BoNT/A vs. Normal saline [36]	13.7	7.6	28	14.7	6.86	31	21.2	9.31	28	19.7	10.6	31
Kasyan, 2012 [39]	24.2	4.6	15	21.9	3.8	17	14.6	13.1	15	26.9	9.8	17
Kuo, 2009, BoNT/A 200 U vs. Hydrodistension [40]	10.2	6.48	15	13.1	5.95	23	11.5	7.26	15	13.6	5.62	23
Kuo, 2009, BoNT/A 100 U vs. Hydrodistension [40]	14.1	6.1	29	13.1	5.95	23	15.1	4.54	29	13.6	5.62	23
Kuo, 2016 [41]	10.7	5.4	40	10.4	3.8	20	12.1	8.6	40	9.9	4.2	20

Results are expressed in mL/s, as mean ± SD for the active and the control agents, at both baseline and follow-up (f-up). * Value has been reported as difference from baseline.

**Table 9 toxins-11-00510-t009:** RCTs on the use of botulinum neurotoxin A intravesical injections in the treatment of patients affected by IC/BPS, assessing changes in post-void residual volume.

PVR	Active Agent, Baseline			Control Agent, Baseline			Active Agent, f-up			Control Agent, f-up		
Study, treatment	Mean	SD	No. of pts	Mean	SD	No. of pts	Mean	SD	No. of pts	Mean	SD	No. of pts
Akyhama, 2015 [35]	43.2	39.3	18	32.4	16.2	16	13 *	43.4	18	13.1 *	28.2	16
Chuang, 2017, BoNT/A vs. Lipotoxin [36]	33.9	55.6	28	52.7	59.7	31	24.7	25.4	28	24.9	27.3	31
Chuang, 2017, BoNT/A vs. Normal saline [36]	33.9	55.6	28	58.6	98	31	24.7	25.4	28	31.2	39.2	31
Kasyan, 2012 [39]	13.1	4.3	15	12.3	3.6	17	23.2	3.3	15	13	2.6	17
Kuo, 2009, BoNT/A 200 U vs. Hydrodistension [40]	13.3	41.2	15	38.7	79.3	23	82.7	155.6	15	30.2	50.5	23
Kuo, 2009, BoNT/A 100 U vs. Hydrodistension [40]	30.4	53.2	29	38.7	79.3	23	66.7	106.5	29	30.2	50.5	23
Kuo, 2016 [41]	22.7	48.2	40	61.8	91	20	86.1	115.3	40	64.7	101.9	20

PVR: post-void residual volume. Results are expressed in mL, as mean ± SD for the active and the control agents, at both baseline and follow-up (f-up). * Value has been reported as difference from baseline.

**Table 10 toxins-11-00510-t010:** Side effects from RCTs on the use of botulinum neurotoxin A intravesical injections in the treatment of patients affected by IC/BPS.

Author, Year, Treatment	Hematuria No. of Patients	Dysuria No. of Patients	Large PVR No. of Patients	Urinary Retention No. of Patients	UTIs No. of Patients	Time to Onset
**Akyhama, 2015, [35]**	1 (all participants)	10 (all participants)	3 (> 100 mL) (all participants)	None	2 (all participants)	Between week 1 and month 3
**Chuang, 2017, [36]** BoNT/A vs. Lipotoxin	None	2 (active group) 1 (control group)	None	None	None	Within 4 weeks
**Chuang, 2017, [36]** BoNT/A vs. Normal Saline	None	1 (control group)	None	None	None	Within 4 weeks
**Gottsch, 2011, [38]**	None	None	None	None	None	Within 3 months
**El-Bahanasy, 2009, [37]**	1 (control group)	3 (active group) 5 (control group)	None	None	1 (active group) 2 (control group)	Immediately after the injection
**Kasyan, 2012, [39]**	Not described	Not described	Not described	Not described	Not described	Not described
**Kuo, 2009, [40]** BoNT/A 200 U vs. Hydrodistension	2 (active group)	7 (active group)	5 (active group)	2 (active group)	3 (active group)	Dysuria: between weeks 4 and 8
**Kuo, 2009, [40]** BoNT/A 100 U vs. Hydrodistension	None	3 (active Group) 1 (control group)	2 (active group)	1 (active group)	None	Dysuria: between weeks 4 and 8
**Kuo, 2016, [41]**	2	16 (active group) 1 (control group)	None	1 (active group)	1 (active group) 1 (control group)	Within the first 2 weeks
**Manning, 2014, [42]**	None	None	None	None	7 (active group) 5 (control group)	At some time after the injection, up to month 3
**Pinto, 2018, [43]**	None	None	None	None	3 (active group) 2 (control group)	Between weeks 4 and 12
**Taha-Rasheed, 2010, [44]**	Not described	Not described	Not described	Not described	Not described	Not described

PVR: post-void urinary residual volume; UTIs: urinary tract infections.

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
