# Peer review of "Botulinum Neurotoxin A Intravesical Injections in Interstitial Cystitis/Bladder Painful Syndrome: A Systematic Review with Meta-Analysis"

_toxins, 2019, doi:10.3390/toxins11090510_

Round 1

Reviewer 1 Report

This review describes the effect of botulinum toxin treatment of interstitial cystitis/bladder painful syndrome. The authors presented a systematic meta-analysis review of the literature using Medline and EMBASE search systems about the effects of BoNT-A intravesically injected in patients with interstitial cystitis/bladder painful syndrome between January 2000 and March 2019.

Generally, the paper is carefully written and gives news and valuable information compared with the review from et al. (Toxins 2016, 8, 201) from 2016. Unfortunately, this paper (and also other comparable reviews) was not cited in the present ms. This is not scientifically correct and must be done.

I have the following concerns that must be taken into account.

General aspects

Please use in instead of Botulinum toxin A – Botulinum neurotoxin A

Please use for abbreviations instead of square brackets – round brackets

Please check all spaces between the numbers and symbols

Special suggestions

Keywords: Please include “ intravesical injection”

Introduction:

Line 27, 32: please add more references;

Line 40, 48, 59: please add references;

Please write out once again, not only in Abstract, “IC/BPS”;

Please add more information about the mechanisms of action of the BoNT-A in pain by IC/BPS. For example, see Chiu et al. (2016)

Chiu B, Tai HC, Chung SD, Birder LA. Botulinum Toxin A for Bladder Pain Syndrome/Interstitial Cystitis. Toxins (Basel). 2016 Jul 1;8(7). pii: E201. doi: 10.3390/toxins8070201, PMID:27376330.

Results:

Line 76, 78, 80, 81: please write out the terms used first time in the ms, not only in the abstract

Table 1 and Table 10: First column: add also “treatment”- Author, year and treatment; please describe all abbreviations below the table in the legends, see for example (Giannantoni et al. 2012) which have very comparable designs.

Giannantoni A, Bini V, Dmochowski R, Hanno P, Nickel JC, Proietti S, Wyndaele JJ. Contemporary management of the painful bladder: a systematic review. Eur Urol. 2012 Jan;61(1):29-53. doi: 10.1016/j.eururo.2011.07.069. Epub 2011 Sep 9, PMID:21920661.

Line: 88, 111: please write out :NIDDK, ESSIC and SM.

Tables 2, 3, 4, 5, 6, 7, 8, 9: First column: instead of “Treatment study”, please change to “Study, treatment”. What about 200U, 100U? Please do it more understandable.

 Please describe all abbreviations below the table in the legends, see for example (Giannantoni et al. 2012).

Figures 1, 2, 3, 4, 5, 6, 7, 8:

Please correct all Figures: do it more carefully and describe all figures exactly in respective legends, like as Giannantoni et al. (2012), the design of which is very comparable.

Side effects: line 188, please say: …  in 10 from 12 trials; Line 192, 202: please use lower case letter for a in “authors”;

Discussion: Line 238: please use other synonyms for “overall”, overall is very often using in MS

I completely missing the comparison with another similar previous systematic reviews of RCTs assessing BoNT-A for IC/BPS. What about Chiu et all. (2016) in “Toxins” or Tirumure et al. 2010 in Int Urogynecol and Rahnama et al. 2017 in Neurourol Urodyn? Please discuss exactly what especially is new or different in the present review? For literature search you used only “Medline and EMBASE”, other reviews have done this using also CINAHl, CENTRAL and also with more keywords. Please, include the comparison into the discussion.

Author Response

Responses to Reviewer 1

Comments and Suggestions for Authors

This review describes the effect of botulinum toxin treatment of interstitial cystitis/bladder painful syndrome. The authors presented a systematic meta-analysis review of the literature using Medline and EMBASE search systems about the effects of BoNT-A intravesically injected in patients with interstitial cystitis/bladder painful syndrome between January 2000 and March 2019.

Generally, the paper is carefully written and gives news and valuable information compared with the review from et al. (Toxins 2016, 8, 201) from 2016. Unfortunately, this paper (and also other comparable reviews) was not cited in the present ms. This is not scientifically correct and must be done.

Response. First of all, thank you for your positive comments on the present review with meta-analysis. Instead, we retain the Reviewer is not correct in posing a so negative judgment regarding the omission of the review of Chiu et al. (published in Toxins 2016). We well know the valuable review from these Authors about the use of botulinum toxin A in bladder pain, but we did not cite it in our manuscript only because it did not include a meta-analysis. Other recent, comparable reviews with meta-analysis have been included in our manuscript (Wang et al., 2016; Shim et al., 2016; Zhang et al., 2017). However, to avoid any misunderstanding, the review of Chiu et al., has been now included in the manuscript, pag 2, line 83, as follows: A greater evidence supporting the use of BoNTs in bladder pain was then reported by Chiu et al. [19], thus highlighting again the promising role of the neurotoxin A in treating patients affected by the disease.

In addition, the review of Chiu et al. has been now cited when reporting the mechanisms of action of BoNT/A into the bladder (pag. 2, lines 67-76), as follows: “With regards to the mechanisms of action of BoNT/A at the level of the urinary bladder, it has been demonstrated that the neurotoxin A is able to inhibit the nociceptive, afferent nervous transmission by acting on several receptors and neurotransmitters involved also in neurogenic inflammation [19]. Indeed, BoNT/A has been demonstrated to inhibit the release of SP from cultured dorsal root ganglia neurons [25] and reduces the levels of CGRP in rat trigeminal ganglia cells [26]. In experimental studies in animals, the neurotoxin A has been observed to reduce hyperalgesia and TRPV1 expression in rats with neuropathic pain [27], and the expression of cyclooxygenase-2 in a rat model of capsaicin-induced prostatitis [28]. In humans, BoNT/A has been demonstrated to reduce the urinary level of NGF in patients affected by “IC/BPS” by inhibiting the action of interleukin-1 [29]”. As noted, five more references have been included (Ref. 25-29).

I have the following concerns that must be taken into account.

General aspects

Please use in instead of Botulinum toxin A – Botulinum neurotoxin A

Please use for abbreviations instead of square brackets – round brackets

Please check all spaces between the numbers and symbols

Response. Thank you. All the requested changes have been applied.

Special suggestions

Keywords: Please include “ intravesical injection”

Response. As suggested, it has been now included among the keywords.

Introduction:

Line 27, 32: please add more references;

Response. According to the request of the Reviewer, 3 more references (Ref. 13, 14 and 15) have been now added (pag. 2, lines 56-7).

Line 40, 48, 59: please add references;

Response. Two more references (Ref. 18 and 19) have been added in the text (pag 2, line 65).

Please write out once again, not only in Abstract, “IC/BPS”;

Response. “IC/BPS” has been now included in the Text.

Please add more information about the mechanisms of action of the BoNT-A in pain by IC/BPS. For example, see Chiu et al. (2016)

Chiu B, Tai HC, Chung SD, Birder LA. Botulinum Toxin A for Bladder Pain Syndrome/Interstitial Cystitis. Toxins (Basel). 2016 Jul 1;8(7). pii: E201. doi: 10.3390/toxins8070201, PMID:27376330.

 Response.  According to the suggestion of the Reviewer, the following sentences about the mechanisms of action of BoNT/A have been now included in the text, pag 2, lines 67-76:

“With regards to the mechanisms of action of BoNT/A at the level of the urinary bladder, it has been demonstrated that the neurotoxin A is able to inhibit the nociceptive, afferent nervous transmission by acting on several receptors and neurotransmitters involved also in neurogenic inflammation [19]. Indeed, BoNT/A has been demonstrated to inhibit the release of SP from cultured dorsal root ganglia neurons [25] and reduces the levels of CGRP in rat trigeminal ganglia cells [26]. In experimental studies in animals, the neurotoxin A has been observed to reduce hyperalgesia and TRPV1 expression in rats with neuropathic pain [27], and the expression of cyclooxygenase-2 in a rat model of capsaicin-induced prostatitis [28]. In humans, BoNT/A was able to reduce the urinary level of NGF in patients affected by “IC/BPS” by inhibiting the action of interleukin-1 [29]”. Five more references have been added (Ref. 25-29).

Results:

Line 76, 78, 80, 81: please write out the terms used first time in the ms, not only in the abstract. Response. It has been done.

Table 1 and Table 10: First column: add also “treatment”- Author, year and treatment; please describe all abbreviations below the table in the legends, see for example (Giannantoni et al. 2012) which have very comparable designs.

Response. As suggested, the term “treatment” has now been added in the first column of Table 1 and Table 10. We also described all the abbreviations in the Legends below the Tables. Titles of each table have been more detailed.  

Giannantoni A, Bini V, Dmochowski R, Hanno P, Nickel JC, Proietti S, Wyndaele JJ. Contemporary management of the painful bladder: a systematic review. Eur Urol. 2012 Jan;61(1):29-53. doi: 10.1016/j.eururo.2011.07.069. Epub 2011 Sep 9, PMID:21920661.

Line: 88, 111: please write out: NIDDK, ESSIC and SM.

Response. Thank you, it has been done.

Tables 2, 3, 4, 5, 6, 7, 8, 9: First column: instead of “Treatment study”, please change to “Study, treatment”. What about 200U, 100U? Please do it more understandable.

Response. Accordingly, we have changed the title of the first column in all the Tables. In order to make better understandable the terms “Kuo 2009, 200 U and 100 U”, we performed the following changes, which have been reported in both tables and graphs: “Kuo, 2009, BoNT/A 200 U vs Hydrodistension”; “Kuo, 2009, BoNT/A 100 U vs Hydrodistension”. The same details were added for the 2 trials of Chuang 2017, as follows: “Chuang, 2017, BoNT/A vs Lipotoxin”; “Chuang, 2017, BoNT/A vs Normal Saline”.

 Please describe all abbreviations below the table in the legends, see for example (Giannantoni et al. 2012).

Response. All the abbreviations have been described below the Tables.

Figures 1, 2, 3, 4, 5, 6, 7, 8:

Please correct all Figures: do it more carefully and describe all figures exactly in respective legends, like as Giannantoni et al. (2012), the design of which is very comparable.

Response. As suggested, it has been done.

Side effects: line 188, please say: …  in 10 from 12 trials; Line 192, 202: please use lower case letter for a in “authors”;

 Response. These have been corrected.

Discussion: Line 238: please use other synonyms for “overall”, overall is very often using in MS.

Response. We have omitted the term “overall” in pag 16, line 308, by changing the beginning of the sentence as follows: With regards to the rate of UTIs…

I completely missing the comparison with another similar previous systematic reviews of RCTs assessing BoNT-A for IC/BPS. What about Chiu et all. (2016) in “Toxins” or Tirumure et al. 2010 in Int Urogynecol and Rahnama et al. 2017 in Neurourol Urodyn? Please discuss exactly what especially is new or different in the present review? For literature search you used only “Medline and EMBASE”, other reviews have done this using also CINAHl, CENTRAL and also with more keywords. Please, include the comparison into the discussion.

Response. As described before, we have now mentioned in the text the Review of Chiu et al (Toxins 2016), reporting their main observation, also with regards to the mechanisms of action of BoNT/A into the bladder (pag 2, lines 67-76). Now it has been cited several times along the text. We want to underline again we did not previously include this review in our manuscript because it misses a meta-analysis.

As suggested by the Reviewer, in the Discussion Section we have now included a comparison with the results of the first published review on this topic by Tirumuru et al., although we previously retained it presents with a very limited number of trials and a comparison with those results is practically impossible.  We think that the suggested retrospective study of Rahnama et al. (Neurourol Urodyn 2017) should not be appropriate to be included in the present review with meta-analysis, as its topic was related to the use of BoNT/A in male patients affected by OAB. In addition, as suggested by the Reviewer, we have also repeated our search using CINAHL, CENTRAL and MetaRegister of Controlled Trials, which have now indicated in the text (see pg. 17, line 361) by also including 3 more search terms: botox, dysport, xeomin (see pag. 17, line 364), with the same results.

In synthesis, in pag 16, lines 320-21, we added the following new sentences:  Zhang and co-workers in their net-work meta-analysis on the intravesical treatment of “IC/BPS” found that BoNT/A had the highest probability to be the best therapy for “IC/BPS” patients, in comparison with other intravesical treatments, as BCG, PPS, lidocaine, chondroitin sulfate and resiniferatoxin [32]. In pag 16, lines 325-333, the following sentences have been now added: Presumably, the larger number of trials with a higher number of treated patients included in our review with meta-analysis, as compared to those of Shim et al., allowed us to detect the significant improvement also in ICSI after BoNT/A intravesical injections. Tirumuru et al., included in their systematic review only 3 RCTs and 7 prospective studies, with a limited number of treated “IC/BPS” patients with the neurotoxin A [33]. They found improvement in symptoms in 2 trials and 6 prospective studies, but meta-analysis was not performed due to a great heterogeneity of the included outcomes, and it was stated that no solid conclusions could derive from their review on the effectiveness of BoNT/A intravesical injection in the treatment of “IC/BPS”. [33].

Reviewer 2 Report

Introduction:

you report no satisfactory results for current treatment modalities for CPPS - what data supports this? Your goal was to evaluate efficacy of botox in patients with IC/CPPS in terms of pain, frequency and capacity.  Are you presuming that patients with IC/CPPS would not have similar efficacy as patients without pain in terms of frequency and capacity?  If so, why?

Methods:

This section should be after the introduction.

Results:

Overall the results section is very difficult to read the text.  The tables and figures are appropriate but the text is not particularly useful as it only describes the studies used and does not summarize anything seen in the tables or figures nor does it highlight any results in the tables or figures.

How did you determine the size of effect - ie small vs. medium?

Discussion:

with such variability in the RCT used in terms of techniques, do you feel that the data is comparable and appropriate to do a meta-analysis?   This seems to be more of a systematic review than a meta-analysis as most of the data is presented as findings in each study as opposed to a true meta-analysis.

Overall this study more summarizes the findings of other studies available and gives little information that is not already available to readers.  Though the topic is lacking in literature, this is not a true meta-analysis as the data is not combined which is the benefit of a meta-analysis.

Author Response

Responses to Reviewer 2.

Introduction:

Questions. you report no satisfactory results for current treatment modalities for CPPS - what data supports this? Your goal was to evaluate efficacy of botox in patients with IC/CPPS in terms of pain, frequency and capacity.  Are you presuming that patients with IC/CPPS would not have similar efficacy as patients without pain in terms of frequency and capacity?  If so, why?

Response to point 1. Unfortunately, the existing literature about treatments’ efficacy for patients affected IC/CPPS shows an overall poor effectiveness. Recently, the MAPP Research Network Study Group stated that: “UCPPS is poorly understood, and treatment is mostly empyrical, with unsatisfactory patient outcomes” (Clemens et al., Nat Rew Urol 2019). In addition, it has been recently affirmed that “Chronic pain commands particular attention: it is usually difficult to treat, particularly certain types such as neuropathic pain” (Dosenovic et al., Anesth Analg. 2017). Taking a cue from these statements, an additional, more recent reference (Ref. 4) has been included at the end of the relative sentence (pag. 1, line 33).

Response to point 2. The aim of the present review with meta-analysis was to investigate the efficacy and safety of BoNT/A intravesical injections in IC/BPS patients who have been refractory to conventional treatments. To do this, we analyzed the outcomes considered in all the available RCTs assessing this issue. Among these, pain, urinary frequency and bladder capacity have been investigated. As results, we observed a significant improvement in pain (medium effect size of SMD for changes in VAS for pain) and a significant improvement also in daytime urinary frequency (medium effect size of SMD for changes in VAS for pain). For what concern functional bladder capacity, we did not identify any significant improvement, and we already underlined this surprising, negative result in the Discussion section, pag 16, line 300, hypothesizing that it could have been affected by the results of 1 trial (Ref. 35). Thus, we do not presume that the neurotoxin A is less effective in improving bladder capacity in IC/BPS patients, as compared to patients without pain. We only report what derives from the analysis of the results.

This point has been now made clearer in pag 16, lines 301-303, with the following sentence: “This presumably avoided to obtain the expected result of a significant improvement in bladder capacity, what is usually observed after BoNT/A injections in patients with overactive bladder. A pertinent reference has been added (Ref. 47).  

Methods:

This section should be after the introduction.

Response. We followed the requirements from the Instructions for the Authors in Toxins, which ask to describe methodology after showing the results.

Results:

Overall the results section is very difficult to read the text.  The tables and figures are appropriate but the text is not particularly useful as it only describes the studies used and does not summarize anything seen in the tables or figures nor does it highlight any results in the tables or figures.

Response. The included Tables (from Table 2 to Table 9) describe the mere results for each outcome in numerical terms (mean, SD, number of patients), deriving from each single trial. Each figure represents the effect size meta-analysis plot (random effect), with the level of significance. As suggested by the Reviewer, in order to summarize the final result for each individual outcome, we have now added in the text detailed explanations of the obtained changes (in pag. 6, line 136; in pag. 8, lines 160-70; pag. 8, and 176-77; in pag. 9, lines 191-93; pag. 10, line 207-08; pag. 11, line 220; pag. 13, line 244-45). We think that many details are reported particularly in pag 7, lines 149-155, but we retain it is important when describing the results, to include the references of the reported studies and to detail treatments’ modalities for each individual study.

How did you determine the size of effect - ie small vs. medium?

Response. As already described in Methods, (pag 17 and pag 18, lines 375-386), according to the study of Cohen J. (Ref. 50) the size of the effect is determined by the mean difference between two variables expressed in standard deviation units. Thus, a SMD of 0.2 standard deviation units is considered a small difference between the intervention groups; a SMD of 0.5 indicates a medium difference, and a value of 0.8 expresses a large difference.

Discussion:

With such variability in the RCT used in terms of techniques, do you feel that the data is comparable and appropriate to do a meta-analysis? This seems to be more of a systematic review than a meta-analysis as most of the data is presented as findings in each study as opposed to a true meta-analysis.

Response.

We understand the perplexity of the Reviewer, which we also had at the beginning of our work. Even though the included trials showed a great variability in terms of techniques, performing a meta-analysis in this review, allowed us to make order about the data coming from the existing literature, and allowed us to detect important results, such as the significant effects of BoNT/A intravesical injections in improving VAS for pain, daily frequency of urination and ICSI in patients with IC/BPS. As mentioned in the text (pag. 17, line 339-42), heterogeneity did not affect ICSI and VAS/Likert scale in the selected trials, what at least confirms it was appropriate to perform a meta-analysis.

Overall this study more summarizes the findings of other studies available and gives little information that is not already available to readers.  Though the topic is lacking in literature, this is not a true meta-analysis as the data is not combined which is the benefit of a meta-analysis.

Response.

We retain the present systematic review with meta-analysis not only summarizes the findings already available, but clearly points out what are the effects of BoNT/A intravesical treatment in this particular disease, to the light of new conducted trials. Previously, only 2 reviews with meta-analysis were performed on the same topic (Wang et al., 2016, Shim et al., 2016), and in only one the same methodology with SMD was used (Shim et al., 2016), but it was applied only in 5 trials. This has been already discussed in the Discussion Section, pag. 16, from lines 313 to line 332.

Although it was not easy, we retain to have performed a true meta-analysis, as we always combined the data of each individual outcome in the respective meta-analysis and the pooled effect sizes are summarized as a graphic synthesis (diamond with 95% confidence limits on the bottom of the graph area), and as a numerical synthesis, on the bottom of each figure. In order to underline the above-mentioned concerns, we added the following sentence in page 17,  lines 342-347: “If on the one hand, performing a meta-analysis could have been inappropriate due to the heterogeneity identified, combining the data of each individual outcome in the respective meta-analysis and summarizing the pooled effect sizes, however helped us in making order about data coming from the existing literature, and allowed us to point up what are the evidences of BoNT/A intradetrusorial treatment in “IC/BPS”.       

Round 2

Reviewer 1 Report

The review in the revised form is massively improved and now I think meets the level required for the journal

Remark: in the abstract say...although not clinically relevant... instead of ...although no clinically relevant...

Reviewer 2 Report

I appreciate the expansive explanations to the prior questions.

The results section is much easier to follow, making the entire manuscript much easier to read.